# Hepatitis C Virus and Hepatitis B Virus Co-Infection

**DOI:** 10.3390/v12070741

**Published:** 2020-07-10

**Authors:** Yi-Fen Shih, Chun-Jen Liu

**Affiliations:** 1Department of Physical Therapy and Assistive Technology, National Yang-Ming University, Taipei 112, Taiwan; yfshih@ym.edu.tw; 2Hepatitis Research Center, National Taiwan University Hospital, Taipei 100, Taiwan; 3Graduate Institute of Clinical Medicine, National Taiwan University College of Medicine, Taipei 100, Taiwan; 4Department of Internal Medicine, National Taiwan University College of Medicine and National Taiwan University Hospital, Taipei 100, Taiwan

**Keywords:** co-infection, hepatitis B virus, hepatitis C virus, pegylated interferon, direct-acting antivirals, reactivation

## Abstract

Hepatitis C virus (HCV) and hepatitis B virus (HBV) co-infection can be encountered in either virus endemic countries. Co-infection can also be found in populations at risk of parenteral transmission. Previous studies demonstrated a high risk of liver disease progression in patients with HCV/HBV co-infection; thus, they should be treated aggressively. Previous evidence recommended therapy combining peginterferon (pegIFN) alfa and ribavirin for co-infected patients with positive HCV RNA. Recent trials further advise using direct-acting antivirals (DAAs) for the clearance of HCV in the co-infected patients. Reactivation of HBV has been observed in patients post-intervention, with higher risks and earlier onset in those having had HCV cured by DAA- versus pegIFN-based therapy. The mechanism of HBV reactivation is an interesting but unsolved puzzle. Our recent study revealed that in vitro HBV replication was suppressed by HCV co-infection; HBV suppression was attenuated when interferon signaling was blocked. In vivo, the HBV viremia, initially suppressed by the presence of HCV super-infection, rebounded following HCV clearance by DAA treatment and was accompanied by a reduced hepatic interferon response. In summary, major achievements in the treatment of HCV/HBV co-infection have been accomplished over the past 20 years. Future clinical trials should address measures to reduce or prevent HBV reactivation post HCV cure.

## 1. Introduction

In hepatitis B virus (HBV) or hepatitis C virus (HCV) endemic countries, patients are exposed to the risk of being co-infected with both viruses [1,2,3]. Parenteral viral transmission could also lead to HCV/HBV co-infection. In patients infected with both HCV and HBV, the risk of developing liver cirrhosis (LC) and hepatocellular carcinoma (HCC) is usually higher than those with mono-infection of either virus [1,2,3,4,5,6,7,8,9]. Therefore, patients co-infected with hepatitis C and B require regular monitoring and aggressive antiviral treatment.

Previous clinical trials suggested that peginterferon (pegIFN) alfa plus ribavirin (RBV) was effective for clearing HCV in patients with HBV/HCV co-infection [10,11,12,13,14]. However, this regimen may not be suitable for patients with decompensated liver cirrhosis or other contraindications. The optimal treatment strategies for co-infected patients who are contraindicated for IFN-based therapy were challenging before the advent of direct-acting antiviral (DAA)-based anti-HCV therapy. The DAA-based therapy has wider indications and increases the rate of HCV clearance in mono-infected patients; being not only safer, but also much more effective than IFN-based therapy. DAA-based therapy shows potential for filling the unmet gap when treating co-infected patients. Our recent multicenter trial in Taiwan has just revealed that the HCV sustained virologic response (SVR) rate was high (100%) in co-infected patients with HCV genotype 1 or 2 infection who received 12-week sofosbuvir/ledipasvir therapy [15].

Patients may experience HBV reactivation after the cure of their HCV by pegIFN- or DAA-based therapy, the risk of HBV reactivation being higher and onset being earlier in the latter, as warned by the US FDA years ago [16]. Our trial data demonstrated that 73% of the patients experienced HBV reactivation events during the 12-week DAA treatment period and 108-week post-treatment observation period [17]. The risk of reactivation was highest during the treatment period and 12% of the patients with HBV reactivation experienced clinical hepatitis activity and required anti-HBV treatment. Moreover, HBV reactivation can be documented in patients with occult HBV infection (anti-hepatitis B core (anti-HBc) total positive, HBsAg negative).

Until now, the mechanism of this HBV reactivation, post-HCV cure, remained an interesting mystery. Our recent study provided evidence that HBV suppression was attenuated when IFN signaling was blocked in vitro [18]. Clinically, HBV viremia, after initial suppression by HCV super-infection, rebounded following HCV clearance by DAA treatment and was accompanied by a reduced hepatic interferon response.

Management of HBV activity, including the prevention of HBV reactivation post HCV cure, is a clinical issue that needs to be dealt with in further clinical trials. The lack of supportive evidence is reflected by the differences among the recommendations by the EASL (European Association for the Study of the Liver), AASLD (American Association for the Study of Liver Diseases) or APASL (Asian Pacific Association for the Study of the Liver) community regarding the prevention of HBV reactivation in co-infected patients [18,19,20].

This article will review recent updates about the clinical outcomes and treatment of patients with hepatitis C and B co-infection, with emphasis on the risks and mechanisms of HBV reactivation.

## 2. Clinical Outcomes and Predictors of HCV/HBV Co-Infection

The adverse effect of HCV/HBV co-infection has been demonstrated previously by a long-term community-based study in Taiwan showing the combined effect of HCV/HBV co-infection on the cumulative incidence of HCC [9]. Our recent hospital-based cohort study echoed the adverse impact of HCV/HBV co-infection on liver disease progression [21,22]. During a 10-year follow-up, 111 co-infected patients had a higher risk of HCC and cirrhosis than 111 patients with HBV mono-infection, with the hazard ratio of 3.6 and 2.5, respectively [22].

The fibrosis 4 parameters (FIB-4) index has been investigated widely for the assessment of the liver fibrosis stage and the clinical outcomes in patients with chronic hepatitis C [21]. To examine the value of FIB-4 in patients with HCV/HBV co-infection, we retrospectively enrolled 152 non-cirrhotic patients with chronic HCV/HBV co-infection: 56 patients receiving pegIFN/RBV therapy and 96 patients being untreated. The association between the FIB-4 index and the incidence of LC and HCC was explored. We found that the FIB-4 index decreased only in the treated group which achieved SVR; high baseline FIB-4 index in the treated groups independently correlated with a higher risk of developing LC and HCC. These findings indicated that co-infected patients with elevated pre-treatment FIB-4, were more susceptible to liver disease progression, and should be closely monitored post-HCV cure. The potential role of FIB-4 was also evaluated by Butt A et al. [23]. Of 115 HCV/HBV co-infected patients treated with DAA, a decreased SVR rate was observed with more severe liver fibrosis, as determined by the FIB-4 score.

Overt HBV infection (indicated by positive serum HBsAg) adversely affects the clinical outcomes of chronic HCV infection [24]. Occult hepatitis B infection (OBI) is not rare in countries that are endemic for HBV and is commonly encountered in patients with chronic HCV infection in these endemic countries. Nevertheless, the role of OBI in the progression of liver diseases in patients with chronic hepatitis C is not fully understood. To explore the contribution of concurrent OBI to the progression of liver diseases, we investigated the value of the total anti-hepatitis B core (anti-HBc) antibody as a surrogate OBI biomarker in 183 patients with active HCV infection (serum HCV RNA positive) and resolved HBV infection (anti-HBc positive and HBsAg negative). OBI was identified using a sensitive commercial assay. The results showed that 56 (30.60%) of these 183 patients with active HCV infection had OBI. The presence of OBI did not correlate with any adverse clinical outcome in multivariate analyses. Our findings supported that OBI infection may not contribute to the development of adverse liver outcomes in patients with chronic HCV infection.

In short, recent studies have reached a consensus that overt, but not occult, HBV co-infection may accelerate the progression of HCV-related liver diseases. FIB-4 index can help to predict adverse clinical outcomes in CHC patients receiving DAA.

## 3. Treatment of Patients with HCV/HBV Co-Infection: PegIFN and RBV

Our previous data confirmed the efficacy of pegIFN plus RBV in the treatment of co-infected patients with active hepatitis C [10,14,25,26,27]. The durability of hepatitis C clearance in HCV/HBV co-infected patients was later demonstrated by a 5-year prospective follow-up study [14]. This leads to the suggestion that the efficacy and durability of HCV SVR by using pegIFN alfa and ribavirin therapy were satisfactory and not influenced by HBV co-infection. In addition to the cure of HCV infection, pegIFN-based therapy may also help control the chronic HBV infection in patients with HCV/HBV co-infection. During a 5-year post-treatment follow-up, the rate of HBsAg seroclearance was 5.4% per year [14]. Another study, which investigated the outcomes in 192 HCV/HBV co-infected patients after anti-HCV therapy, also found that 67 (34.9%) patients had a favorable outcome of HBsAg seroclearance [28]. The probability was 5.7 per 100 person years. In addition, a pretreatment HBV DNA level of 300 IU/mL served as an independent predictor for the outcome.

For patients with HCV/HBV co-infection, we hope treatment can decrease the risk of HCC development and liver-related mortality in the long term. In Taiwan, we analyzed nationwide databases and conducted a case-control study. We found that the treatment of co-infected patients using pegIFN plus RBV indeed reduced risk of HCC development and liver-related mortality [29].

## 4. DAA-Based Therapy

With the arrival of the DAA regimen, which has proven to be a simple, time-saving, highly tolerated and effective treatment option for HCV [30], two questions arise. First, whether DAA-based therapy has a higher efficacy than pegIFN plus RBV in the seroclearance of HCV RNA, and second, whether co-existing HBV infection influences the response to DAA-based therapy when managing co-infected patients. It is also very important to understand the risk and timing of HBV reactivation after the seroclearance of HCV RNA.

To answer these questions, we conducted a multi-center study in Taiwan using sofosbuvir/ledipasvir to treat HCV/HBV co-infected patients [15]. Overall 111 patients were enrolled (61% HCV genotype 1% and 39% genotype 2; 16% compensated cirrhosis). All patients (100%, 108/108) achieved HCV SVR. The SVR was durable for 108 weeks after the end of the DAA therapy. Our data suggested that a DAA regimen such as 12-week ledipasvir/sofosbuvir was highly effective in patients with HCV co-infected with HBV. The treatment response would not be influenced by co-existing HBV infection.

Unlike the HCV treatment response, the outcomes of co-existing HBV infection were still unclear. One study followed the serial HBsAg and HBV DNA levels in 79 HCV/HBV co-infected patients receiving DAAs (13 receiving anti-HBV nucleos(t)ide analogue [NUC] therapy simultaneously) [31]. The authors found that DAA-treated HCV/HBV co-infected patients had significantly higher rates of HBV seroclearance, in addition to a lower risk of HBV reactivation among those with a low pre-treatment HBsAg titer. Notably, severe HBV-related clinical reactivation can occur in patients not receiving NUC therapy. Of the 79 patients, 6 patients experienced HBV clinical reactivation. Four of the six patients were cirrhotic, and three of the four cirrhotic patients developed reactivation-related liver failure. Their findings strongly implied that anti-HBV NUC prophylaxis should be considered before the start of DAA for HCV in co-infected patients. 

A recent study reported on whether the clearance of HCV by DAA could improve the clinical outcomes of co-infected patients [32]. For persons with baseline significant fibrosis, as indicated by a FIB-4 score of 1.46–3.25, cirrhosis incidence was 49.3/1000 patient-years among 151 HCV/HBV co-infected patients and 18.2 among HCV mono-infected patients (*p* = 0.03). Cirrhosis risk was numerically higher among HBV/HCV co-infected patients than among HCV mono-infected patients, but became lower among those who attained SVR (HR: 0.52; 95% CI: 0.42–0.63).

## 5. Risk and Management of HBV Reactivation Post HCV Cure Risk of HBV Reactivation in the Era of DAA

HBV activity may reactivate during anti-HCV therapy in co-infected patients [33]. Our prior data demonstrated that by using pegIFN plus RBV to treat co-infected patients, reactivation of HBV DNA was found in 61.8% of patients with low pre-treatment serum HBV DNA level. The reactivation may occur either during or after the end of IFN-based therapy [14].

After the introduction of DAA for the treatment of chronic hepatitis C, there is an increased awareness of HBV reactivation in CHC patients co-infected with HBV treated with DAAs. An earlier systematic review and meta-analysis compared the rate of HBV reactivation in CHC patients co-infected with HBV treated with IFN-based therapy, versus those with DAAs [34]. Overall, the pooled incidence rate of HBV reactivation among CHC patients with HBV co-infection (*n* = 779) was similar between those treated with IFN-based therapy (14.5%, *p* < 0.001) and DAAs (12.2%, *p* = 0.03). Interestingly, HBV reactivation was noted to occur much earlier in those treated with DAAs in comparison to those receiving IFN-based therapies. The risk of hepatitis due to HBV reactivation was also higher in those receiving DAA versus IFN-based therapy (12.2% vs. 0%).

To clarify the chronological profile and risk of HBV reactivation during and after the DAA treatment, we followed the HBV virological parameters prospectively for 108 weeks following the end of treatment [17]. We found that during the 108 weeks after treatment, 81 (73%) of the 111 co-infected patients experienced HBV virologic reactivation; which developed most commonly before week 12 of the follow-up period (86%, 70/81). Clinical reactivation occurred in 10 (9%) of the 111 patients. Notably, clinical reactivation can occur late; four patients experienced clinical reactivation between weeks 12 and 48 of the follow-up period. Our findings demonstrate that HBV reactivation can develop in the majority of HCV/HBV co-infected patients treated with DAAs for HCV. Most patients were asymptomatic with HBV virologic reactivation; only a small group required HBV treatment. It has to be noted that clinical reactivation may still occur >3 months after the end of therapy.

In addition to our trial, other studies also reported the risk of HBV reactivation (defined by an increase in serum HBV DNA ≥1 log_10_ IU/mL) ranging from 25% to 87.5% (mean: 41.1%) in HBsAg-positive patients treated by DAAs (Table 1). These findings send a clear message that the risk of HBV reactivation was high in co-infected patients. The incidence of severe hepatitis activity can be minimized through regular monitoring of serum HBV DNA and prompt administration of anti-HBV NUC upon HBV reactivation, as done in our clinical trial.

## 6. Viral Interactions: With Emphasis on Mechanisms of HBV Reactivation

Although both hepatitis B and hepatitis C viruses are hepatotropic viruses whose primary replication occurs in the liver, significant differences exist between these two viruses regarding immune responses after acute and chronic infection [42]. When both viruses exist in the same liver, the situation may be even more complicated. In HBV endemic areas, most of the HCV/HBV co-infected patients have similar HCV RNA levels as patients with HCV mono-infection, but relatively low levels of serum HBV DNA compared with patients with chronic HBV mono-infection [43,44,45,46]. This suggests that the interference between the two viruses is more likely characterized by an inhibition of HBV, exerted by HCV. Because of the frequently encountered problems related to HBV reactivation post HCV cure, understanding the interactions between HCV and HBV may provide clues for future development of prophylactic or therapeutic strategies.

A recent study has investigated the impact of co-existing HBV on circulating T follicular helper cell (Tfh) distribution and the HCV neutralizing antibody response [47]. Patients with HCV mono-infection (*n* = 83) and HBV/HCV co-infection (*n* = 78) were both enrolled. The authors found that compared with HCV mono-infection, the HBV/HCV co-infection group showed significantly lower HCV neutralizing antibody responses and a decreased frequency of circulating Th1-like Tfh cells. The clinical implications of these immunologic interactions between the two viruses await further study.

To further explore the molecular mechanisms behind this viral interaction and resultant HBV reactivation, our recent in vitro and in vivo investigations were conducted with major inputs from virologists [18]. In co-infected cell culture and humanized mice, HBV replication was suppressed by HCV co-infection. In vitro, HBV suppression was attenuated when interferon signaling was blocked. In vivo, HBV viremia, after initial suppression by HCV superinfection, rebounded following HCV clearance by DAA treatment that was accompanied by a reduced hepatic interferon response. Using blood samples of co-infected patients, IFN-stimulated gene products including C-X-C motif chemokine 10 (CXCL10), C-C motif chemokine ligand 5 (CCL5), and ALT were identified to have predictive value for HBV reactivation after HCV clearance. Taken together, our data suggest that HBV reactivation is a result of diminished hepatic interferon response following HCV clearance and furthermore, we have identified serologic markers that can predict HBV reactivation in DAA-treated HBV/HCV co-infected persons.

Another study investigated potential mechanism of HBV reactivation after HCV elimination by DAA treatment in HCV/HBV co-infected patients [48]. Murai et al. examined RIG-I-like helicase (RLH) pathway activation by HCV/HBV co-infection and interference between HBV and HCV in primary human hepatocytes. They found that HCV infection activated the RLH pathway and suppressed HBV replication. Elimination of HCV, by DAA administration, downregulated the RLH pathway and upregulated HBV replication in mice. These findings partly explained the upregulation of HBV replication after HCV elimination, probably through RLH pathway down-regulation.

In short, recent studies have provided evidence regarding the in vivo and in vitro interactions between HCV and HBV. These interactions partly explained why HBV reactivates after the clearance of HCV. However, more studies are needed to fully clarify this issue.

## 7. Updated International Guidelines/Guidance on the Treatment of Patients with HCV and HBV Co-Infection

EASL guidelines [19] suggest that we can treat patients with HCV/HBV co-infection by using the same anti-HCV regimens as in HCV mono-infected patients. Co-infected patients fulfilling the criteria for HBV treatment should be treated according to the EASL 2017 Guidelines on HBV. For co-infected patients who are HBsAg-positive, NUC prophylaxis is recommended before the start of DAA therapy, till at least week 12 post DAA therapy (Table 2). HBV activity should be monitored after stopping HBV treatment.

AASLD practice guidance [20] has different recommendations about the NUC prophylaxis in co-infected patients. Determination of HBV treatment should be based on serum HBV DNA and ALT levels as per the AASLD HBV guidelines for HBV mono-infected patients. For co-infected patients positive for HBsAg, significant risk of HBV DNA and ALT flares exists during DAA therapy for HCV. Thus, HBV DNA levels should be monitored every 4 to 8 weeks during DAA treatment and for 3 months post-DAA treatment in those not receiving NUC therapy before the start of DAA. The risk of HBV reactivation in HBsAg-negative, anti-HBc positive patients is very low during HCV DAA therapy. HBV DNA and HBsAg testing is reserved only for those patients who experience ALT flares during DAA therapy.

Recent APASL guidelines suggest that for these co-infected patients, careful attention should be paid to HBV reactivation for 24 weeks post-treatment [49].

All guidelines agree that the status of HBV infection should be monitored closely if IFN-free regimens are adopted, since HCV DAA is anticipated to have no direct or immunomodulatory effect on the replication of HBV. Anti-HBV therapy should be timely implemented if clinically indicated (Table 3).

Despite being suggested by EASL guidelines, whether prophylactic NUC would be beneficial in comparison to the therapeutic strategy for HBV reactivation remains unclear. We therefore conducted a prospective multicenter trial in Taiwan investigating the effect of prophylactic NUC on the prevention of HBV reactivation after the start of DAA for CHC [NCT04405011]. The data will be available soon and will further the understanding and management of HCV/HBV co-infection.

## 8. Summary and Future Directions

HCV/HBV co-infection is commonly encountered in endemic areas and among individuals at risk of parenterally transmissible infections. How the two viruses interact with each other in the liver is partly understood. In addition to academic interest, these data will help develop more effective antiviral therapy and efficient strategies to prevent HBV reactivation post HCV cure by DAA.

For HCV/HBV co-infected patients, the same genotype-dependent treatment recommendations for single chronic hepatitis C can be applied. The value of DAA-based therapy in co-infected patients has been demonstrated in a recent clinical trial and prolonged follow-up [15]. The long-term benefits of curing HCV by DAAs await further observatory studies. However, HBV reactivation remains a concern to be resolved. Last but not least, for patients with active hepatitis B, more studies are needed to determine the optimal regimen to simultaneously treat both viruses.

## Figures and Tables

**Table 1 viruses-12-00741-t001:** Risk of hepatitis B virus (HBV) reactivation in hepatitis C virus (HCV)/HBV co-infected patients receiving direct-acting antiviral (DAA) therapy for chronic hepatitis C.

Authors (Year)	Clinical Setting	Patient Number	Risk of HBV Reactivation, Number (%)	Outcome
Gane et al. [35] (2016)	Trial	8	7 (87.5)	None associated with clinical HBV flares
Doi et al. [36] (2017)	RWD	4	2 (50)	
Kawagishi et al. [37] (2017)	RWD	4	2 (50)	None associated with clinical HBV flares
Liu et al. [15] (2018)	Trial	111	70 (63.1)	Clinical hepatitis in 5
Tamori et al. [38] (2018)	RWD	22	3 (13.6)	Baseline HBV DNA level <2000 IU/mL in all. Hepatitis flare did not occur in these 3 patients.
Wang et al. [39] (2017)	RWD	10	3 (30)	Clinical hepatitis in 3 (1 with jaundice and 1 with liver failure)
Mucke et al. [40] (2017)	RWD	9	5 (55.6)	NUC therapy for HBV reactivation in 3
Yeh et al. [31] (2020)	RWD	79	30 (38.0)	-11-month post-DAA follow-up -13 received NUC simultaneously -6 (including 4 cirrhotics) developed clinical hepatitis; -3 of the 4 cirrhotics developed liver failure and 2 died despite immediate NUC therapy
Londono et al. [41] (2017)	RWD, prospective	10	5 (50)	None associated with clinical hepatitis activity

Coinfection indicated by positivity for HCV RNA and HBsAg in the serum; DAA, direct-acting antiviral; RWD, real world data; NUC, nucleos(t)ide analogue. Definition of HBV reactivation differed in different studies: Either increases in serum HBV DNA or HBV DNA reappearance.

**Table 2 viruses-12-00741-t002:** International guidelines for management of HBV reactivation in HCV/HBV co-infected patients receiving DAA therapy for chronic hepatitis C.

	AASLD (2018)	EASL (2017)	APASL (2019)
Test for HBsAg in patients with CHC	Yes	Yes	Yes, in HBV high endemic areas
Prophylactic use of NUC at the start of DAA	No, monitor HBV DNA and ALT every 4–8 weeks and for 3 months post-DAA	Yes, concomitant NUC prophylaxis until week 12 post-DAA	-Indicated for patients with advanced fibrosis, cirrhosis or previous HCC. -For patients without the above indications, prophylactic use of NUC or close monitoring is recommended. -Follow through 24 weeks after end of DAA.
Treatment of chronic hepatitis B	Per AASLD guidelines	Per EASL guidelines	Stopping rule per APASL guidelines
Remark			

AASLD, American Association for the Study of Liver Diseases; EASL, European Association for the Study of the Liver; APASL, Asian Pacific Association for the Study of the Liver; CHC, chronic hepatitis C; DAA, direct-acting antiviral; NUC, nucleos(t)ide analogue.

**Table 3 viruses-12-00741-t003:** Proposed strategies for the treatment of patients with HCV/HBV co-infection.

HCV RNA Positivity	HBV DNA Level	Treatment Goals	Proposed Strategies	Remarks
Detectable	<2000 IU/mL	Cure of HCV infection	DAA *	HBV reactivation is a concern. Prophylactic or therapeutic NUC per regional guidelines. Prophylactic NUC is suggested in co-infected patients with advanced fibrosis or cirrhosis.
Detectable	≥2000 IU/mL	Cure of HCV infection; evaluating control of HBV replication	DAA * Or DAA + NUC	HBV reactivation is a concern. Per regional guidelines for treatment of chronic hepatitis B. AASLD, APASL and EASL have different recommendations about prophylactic NUC. Prophylactic NUC is suggested in co-infected patients with advanced fibrosis or cirrhosis.
Undetectable	≥2000 IU/mL	Control of HBV replication	NUC or pegIFN	Per regional guidelines for treatment of chronic hepatitis B.
Undetectable	<2000 IU/mL	None	Clinical observation	Per regional guidelines for treatment of chronic hepatitis B.

PegIFN, peginterferon; NUC, nucleos(t)ide analogue; DAA, direct acting anti-viral; HCV, hepatitis C virus; HBV, hepatitis B virus. * Data from large multicenter clinical trial.

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
