# Peer review of "Hepatitis C Virus and Hepatitis B Virus Co-Infection"

_viruses, 2020, doi:10.3390/v12070741_

Round 1

Reviewer 1 Report

In introduction it should be additionally emphasised that DAA schedules are not only safer but much more effective than IFN-based schedules in HCV clearance

Rows 49-50 - please, clear that the high rate of HBV reactivation was observed in HBsAg positive patients, as also in occult HBV infection (anti-HBc total positive, HBsAg negative) the cases of HBV reactivation were noted post-DAA treatment.

Rows 119-120 - please, check the evidence for the efficacy of pegIFN/RBV treatment - it is not true that it is similar to the DAA schedules (e.g. in genotype 1 HCV infections SVR rate was about 50-60% in different studies and populations)

row 68 - should be "HCV/HBV"

Author Response

Comment 1:

In introduction it should be additionally emphasized that DAA schedules are not only safer but much more effective than IFN-based schedules in HCV clearance

Response: Thanks for the suggestion. We add relevant background introduction (lines 42-43).

Comment 2:

Rows 49-50 - please, clear that the high rate of HBV reactivation was observed in HBsAg positive patients, as also in occult HBV infection (anti-HBc total positive, HBsAg negative) the cases of HBV reactivation were noted post-DAA treatment.

Response: We add relevant description (lines 54-55).

Comment 3:

Rows 119-120 - please, check the evidence for the efficacy of pegIFN/RBV treatment - it is not true that it is similar to the DAA schedules (e.g. in genotype 1 HCV infections SVR rate was about 50-60% in different studies and populations)

Response: We modify the description (line 122)

Comment 4:

row 68 - should be "HCV/HBV"

Response: We correct the typos (line 70)

Reviewer 2 Report

This is a very nicely written review article written by the real experts in the field. The authors have their own original data on this important topic.

Comments:

1. Does HBV genotype matters in the risk of reactivation after DAA treatment?

2. Does chemotherapy and steroid exacerbate HCV reactivation on top of HBV reactivation?

3. Minor point - Occult hepatitis B infection more often abbreviated as OBI instead of OHB.

Author Response

This is a very nicely written review article written by the real experts in the field. The authors have their own original data on this important topic.

Response: We thank the Reviewer 2’s encouragement.

Comment 1: Does HBV genotype matters in the risk of reactivation after DAA treatment?

Response: We do not have relevant information or research data. We hope to address this issue in future studies.

Comment 2: Does chemotherapy and steroid exacerbate HCV reactivation on top of HBV reactivation?

Response: We also do not have this research data. We hope to address this issue in future studies.

Comment 3: Minor point - Occult hepatitis B infection more often abbreviated as OBI instead of OHB.

Response: We accept this suggestion and modify relevant text content (lines 88-98).

Reviewer 3 Report

From the title of the here presented review article by Shih and Liu, one would expect a comprehensive overview of “Hepatitis C Virus and Hepatitis B Virus Co-infection”. In many parts, the authors yet focus too much on their own recently published studies on this topic and miss to set their findings into the context of up-to-date literature. Especially the parts 2-5 read like an extension/repetition of results parts of primary data.

The superiority of DAA regimen to treat HCV mono-infection, but also co-infections with HBV is long accepted in the field. The authors pay too much attention to IFN-based intervention strategies in my opinion.

This review unfortunately does not add new input and/or knowledge to the field. In fact, most of the summarized findings have already been implemented in the current guidelines of EASL, APASL, and AASLD, as the authors state themselves in part 7.

I agree with the authors, that how the two viruses interact with each other in the liver is only partly understood and that more effective antiviral therapy and efficient strategies to prevent HBV reactivation can only be achieved by further research. Here, I would like to see the focus of this article. How do the authors think, this could be achieved? What do they speculate? I think this is more interesting to give some new input, rather than summarizing studies in detail.

Author Response

We thank the Reviewer 3 for his critical comments and suggestions. We prepare the review article based on cutting edge knowledge and our research findings regarding HCV/HBV co-infection. We hope our future review article may address the concept and focus of the Reviewer.